# When Do Hallucinations Arise? A Graph Perspective on the Evolution of Path Reuse and Path Compression

Xinnan Dai [1]  Kai Yang [2]  Cheng Luo [3]  Shenglai Zeng [1]  Kai Guo [1]  Jiliang Tang [1]

## Abstract

Reasoning hallucinations in large language models (LLMs) often appear as fluent yet unsupported conclusions that violate either the given context or underlying factual knowledge. Although such failures are widely observed, the mechanisms by which decoder-only Transformers produce them remain poorly understood. We model next-token prediction as a graph search process over an underlying graph, where entities correspond to nodes and learned transitions form edges. From this perspective, contextual reasoning is a constrained search over a sampled subgraph (intrinsic reasoning), while context-free queries rely on memorized structures in the underlying graph (extrinsic reasoning). We show that reasoning hallucinations arise from two fundamental mechanisms: **Path Reuse**, where memorized knowledge overrides contextual constraints during early training, and **Path Compression**, where frequently traversed multi-step paths collapse into shortcut edges in later training. Together, these mechanisms provide a unified explanation for reasoning hallucinations in LLMs and connected to well-known behaviors observed in downstream applications. [1]

## 1. Introduction

Large language models (LLMs) have demonstrated remarkable capabilities in reasoning, planning, and decision-making across a wide range of tasks (Naveed et al., 2025; Chang et al., 2024; Yao et al., 2024). However, despite their empirical success, LLMs are known to suffer from

[1]Michigan State University [2]University of Michigan [3]Independent Researcher. Correspondence to: Kai Guo <guokai1@msu.edu>.

*Proceedings of the 43rd International Conference on Machine Learning*, Seoul, South Korea. PMLR 306, 2026. Copyright 2026 by the author(s).

[1]Code is available in the https://github.com/DDigimon/graph_hallucination.

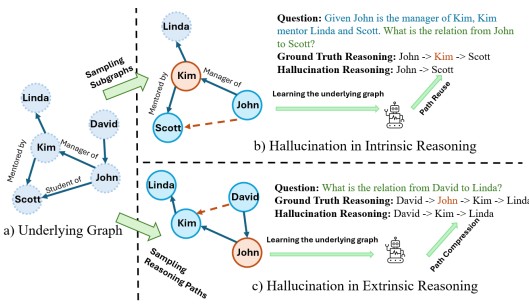

*Figure 1.* Reasoning Hallucinations from Underlying Graph Structures. (a) Building up the underlying graph for the implicit knowledge structure (b) In intrinsic reasoning, the model reuses common paths and hallucinates a direct relation (John → Scott) instead of the correct reasoning provided in context (John → Kim → Scott). (c) In extrinsic reasoning, long reasoning chains (David → John → Kim → Scott) are compressed into shorter paths (David → Kim → Scott), causing hallucinated relations via path compression.

hallucinations, where they generate fluent but incorrect or unsupported content (Huang et al., 2025; Sriramanan et al., 2024; Rawte et al., 2023). To be more specific, hallucinations can be categorized as input-conflict or fact-conflict (Ji et al., 2023). From an intrinsic reasoning perspective, these occur when the model's predictions do not align with the given context. From an extrinsic reasoning perspective, they arise when the outputs are inconsistent with real-world knowledge (Zhang et al., 2025). Understanding why hallucinations arise in such settings, rather than merely how often they occur, remains a central challenge for trustworthy and reliable LLMs.

Existing studies primarily analyze hallucinations from the perspectives of training data distributions and model outputs, attributing hallucinations to mismatches between the model's reasoning trajectories and either the pretraining data or the provided context (Bang et al., 2025; Liu et al., 2026; Kalai et al., 2025). While these analyses capture important surface-level phenomena, they lack a unified explanatory framework that connects hallucinations to the internal reasoning mechanism of decoder-only Transformers. In particular, as prior work (Orgad et al.; Zhu et al., 2024) focuses on models at a fixed checkpoint, it remains unclear when and why hallucinations emerge from the next-token prediction formulation that underlies decoder-only architectures.

To bridge this gap, following prior work that examines the capabilities of decoder-only Transformers (Wang et al., 2025; 2024), we model next-token prediction as a graph search problem within a constrained domain. Motivated by the fact that knowledge is expressed through language sequences (Hogan et al., 2021), we assume that such knowledge is organized over an underlying graph, as illustrated in Figure 1a. In this graph, nodes correspond to entities (illustrated here using person names), while edges represent relations between entities, which are assumed to be invariant to query formulation. Under this view, a context containing relational hints can be regarded as sampling a subgraph from the underlying graph, whereas a direct query without context requires the model to reason along an internal path encoded in its parameters. These two settings naturally correspond to intrinsic reasoning and extrinsic reasoning, respectively.

We then characterize hallucinations in decoder-only Transformers under the underlying graph formulation. As illustrated in Figure 1b, when the context specifies the relations John → Kim → Scott, the model is expected to predict relations that are consistent with this contextual subgraph. However, a hallucination occurs when the model directly predicts John → Scott, which is not supported by the given context. This failure reflects a lack of faithfulness to the provided context, corresponding to intrinsic reasoning hallucination. Meanwhile, as shown in Figure 1c, even when the underlying knowledge structure follows David → John → Kim, the model may bypass John and directly predict David → Kim. This behavior constitutes a hallucination with respect to the underlying knowledge graph, revealing a failure in extrinsic reasoning. Together, these two cases highlight how hallucinations arise from deviations in graph-based reasoning paths, either by violating contextual constraints or by compressing multi-step transitions in the underlying knowledge structure.

Building on this perspective, we formalize the reasoning search space induced by underlying graphs to model both intrinsic and extrinsic reasoning in Section 3. We then identify two core mechanisms underlying reasoning hallucinations in decoder-only Transformers: **Path Reuse** (Section 4) and **Path Compression** (Section 5). Path Reuse arises in early training, when the model learns the underlying graph structure before learning to use contextual constraints. As a result, it reuses memorized reasoning paths, ignoring context-specific information. Path Compression emerges in later training. Oversampling frequent paths biases the learned graph toward high-degree nodes, suppressing low-resource nodes and collapsing multi-step reasoning into shortcuts. We show that both behaviors naturally emerge during pretraining, driven by structural biases in next-token prediction formulation.

We further discuss the implications of these hallucinations for downstream LLM applications in Section 6. From the perspective of the pretraining stage, we analyze how hallucinations naturally emerge when models learn knowledge from language sequences, where structural biases in next-token prediction shape the underlying reasoning graph. From the post-training stage, we examine how different fine-tuning strategies influence the recovery from hallucinations, and why certain compressed or missing reasoning paths are difficult to restore once overwritten. Finally, from the perspective of understanding large language model capabilities, we connect hallucinations to well-known LLM behaviors, such as the reversal curse and reasoning distribution bias, and reinterpret them through the lens of reasoning on the underlying graph.

In conclusion, our contributions are summarized:

1. We develop a unified graph-based perspective to explain the search behavior of Transformers, analyzing how intrinsic and extrinsic reasoning emerge under different underlying graph properties.

2. We further provide an explanation of hallucinations in decoder-only Transformers, summarizing them into two fundamental behaviors: path reuse and path compression. We study when these hallucinations become pronounced, particularly during the pretraining stage.

3. We analyze how reasoning hallucinations arise during pretraining and how post-training fine-tuning affects their recovery. We further connect these hallucinations to well-known LLM behaviors, through a unified graph-based reasoning view.

## 2. Related work

**Synthetic Graphs for LLMs** Graphs have been widely adopted as a principled abstraction for modeling step-by-step reasoning processes in language models (Prystawski et al., 2023; Khona et al., 2024). Building on this perspective, synthetic graph-based search problems are commonly used as controlled testbeds to probe the reasoning and planning capabilities of large language models. Empirical studies show that autoregressive Transformers can partially capture local graph structure from the provided context and exhibit search-like behaviors. However, their performance degrades sharply as graph size or reasoning depth increases, and simply scaling model size or training data does not resolve these failures (Wang et al., 2024; Saparov et al.). Bachmann & Nagarajan (2024) further identifies inherent limitations of next-token prediction in autoregressive Transformers, showing that models are prone to shortcut solutions that utilize prefix tokens rather than performing faithful multi-step reasoning. From a planning perspective, reinforcement learning can improve performance along

selected trajectories, but it often biases models toward a narrow subset of high-reward paths, rather than recovering the full underlying reasoning structure (Wang et al., 2025).

Our work provides a unified framework for understanding hallucinations under training dynamics, covering both context-conditioned and context-free reasoning settings. Moreover, we establish a principled connection between reasoning on synthetic graphs and knowledge acquisition from real-world language sequences.

**Training Dynamics of LLMs**    Understanding how reasoning capabilities emerge and evolve during training is crucial to interpreting model behavior and hallucinations. Recent studies on training dynamics have shown that phase transitions (Wei et al.) occur when specific abilities emerge during training. Research on grokking (Power et al., 2022) shows that models can exhibit delayed generalization by memorizing training data before discovering the underlying patterns. Recent work has investigated the evolution of internal representations during training models (Nanda et al.), revealing that early training stages may prioritize different computational strategies than later stages. This training dynamics results in a divergence between optimization and learning (Mousavi et al., 2026), characterized by a reduction in training loss that decreases without improving the model's capabilities.

Our work advances this understanding by identifying two unique training paradigms of underfitting (Path Reuse) and overfitting (Path Compression) which correspond to hallucinations. Using a purely synthetic graph setting, we further extend this explanation to knowledge learning from language sequences, showing how hallucinations can arise even as training loss continues to decrease.

## 3. Underlying Graph for the Searching Space

Since LLMs perform reasoning through multiple sequential steps, we model the reasoning search space as an underlying directed graph, where nodes correspond to intermediate reasoning states and edges represent valid reasoning transitions between them.

**Definition 3.1** (Underlying Graph). Let $G = (V, E)$ be a directed graph. Each node $v \in V$ denotes an atomic reasoning state (e.g., an entity, a symbolic variable, or an intermediate concept), and each directed edge $(u, v) \in E$ represents a *valid reasoning transition* from state $u$ to state $v$, as permitted by the underlying task semantics or logical constraints.

**Definition 3.2** (Valid Reasoning Path). Given an underlying graph $G = (V, E)$, a *reasoning path* from a source state $s \in V$ to a target state $t \in V$ is a sequence of nodes $p = (v_0, v_1, \ldots, v_L)$, where $v_0 = s$, $v_L = t$, and $(v_{i-1}, v_i) \in E$ for all $i = 1, \ldots, L$. Each path corresponds to a valid multi-

step reasoning process that incrementally transforms the source state into the target state.

Under this formulation, we obtain several advantages. (1) Intrinsic and extrinsic reasoning can be naturally distinguished by whether the reasoning is supported by an explicit subgraph in Section 3.1. (2) Both reasoning settings allow us to enumerate the maximal set of valid reasoning paths. As a result, our experiments can simulate an idealized scenario in which all possible solution paths are available in Section 3.2.

### 3.1. Intrinsic and Extrinsic Reasoning

**Intrinsic Reasoning.**    Intrinsic reasoning requires LLMs to perform reasoning that is strictly aligned with the given conditions. In our setting, the context is specified by a subgraph $G_I$, which is represented as an edge list and denoted by $\mathsf{EL}(G_I)$ (Cohen et al.; Dai et al.). Given a source node $s$ and a target node $t$, a training sample is constructed as $\mathsf{PATH}_I(s, t, G_I) = (\mathsf{EL}(G_I), \mathtt{S}, s, \mathtt{T}, t, \mathtt{PATH}, s, a, b, t)$, where the corresponding query is denoted by $q(G_I, s, t)$. From the subgraph perspective, the context provides all of the relevant edges for the path reasoning.

**Extrinsic Reasoning.**    In contrast, extrinsic reasoning relies solely on the model's memorized knowledge rather than the provided context. The training example is given by $\mathsf{PATH}_C(s, t) = (\mathtt{S}, s, \mathtt{T}, t, \mathtt{PATH}, s, a, b, t)$, where the query is specified only by the source and target nodes as $q(s, t)$. In this setting, prediction is therefore driven primarily by knowledge encoded in the model parameters.

### 3.2. Maximal Settings for Experiments

**Maximum Number of Subgraphs**    The maximum number of training samples for intrinsic reasoning is determined by the number of distinct subgraphs that can be obtained from the underlying graph $G = (V, E)$. Formally, a subgraph $G_I$ is specified by a node subset $U \subseteq V$ together with an edge subset $F \subseteq E[U]$, where $E[U]$ denotes the set of edges in $G$ whose endpoints are both contained in $U$, Let $|V| = N$ and $|E| = M$. Since $0 \leq |E[U]| \leq M$ for any $U \subseteq V$, we have $\#\mathcal{G}_I(G) = \sum_{U \subseteq V} 2^{|E[U]|}$ Notably, the number of subgraphs can grow explosively as graph density increases—for instance, in a fully connected directed graph, this quantity can be $2^{N^2 - N}$. The upper bound represents the maximal combinatorial space when both nodes and edges are freely selected. Consequently, the sample number of intrinsic-reasoning dataset grows jointly with the number of nodes and the edge density of the underlying graph. Each subgraph $G_I$ further induces a family of intrinsic-reasoning queries $q(G_I, s, t)$, and thus the maximal number of intrinsic-reasoning samples is upper-bounded by $\#\mathcal{G}_{\mathrm{sub}}(G)$. Meanwhile, selected subgraphs should preserve multiple valid paths for diverse query–answer pairs.

**Maximal Subgraph Space.** The maximal path space denotes the total number of feasible shortest paths between a source–target pair, which determines the upper bound of extrinsic-reasoning samples. In the extrinsic reasoning setting, the model is queried only with the source and target nodes $(s, t)$ and must therefore rely on memorized transitions rather than explicit contextual constraints. Since source–target pairs are drawn from a finite set of size $|V|^2$, we analyze the number of possible reasoning paths by fixing a given pair $(s, t)$. Let $d = \text{dist}(s, t)$ denote the shortest-path distance between $s$ and $t$. We define the shortest-path layers as $V_i = \{v \mid \text{dist}(s, v) = i, \text{dist}(v, t) = d - i\}, \quad i = 1, \ldots, d - 1$, with boundary layers $V_0 = \{s\}$ and $V_d = \{t\}$. Any shortest path from $s$ to $t$ selects exactly one node from each intermediate layer, yielding a Cartesian-product upper bound $\#\text{SP}(s, t) \leq \prod_{i=1}^{d-1} |V_i|$. However, not all such combinations correspond to valid paths, as reachable path depends on the availability of edges between adjacent layers. Let $\rho_i \in [0, 1]$ denote the edge density between layers $V_i$ and $V_{i+1}$. Observing that $\prod_{i=1}^{d-1} |V_i| = \prod_{i=0}^{d-1} \sqrt{|V_i| |V_{i+1}|}$, the number of realizable shortest paths is therefore bounded by $\#\text{SP}(s, t) \leq \prod_{i=0}^{d-1} \sqrt{\rho_i |V_i| |V_{i+1}|}$. Note that $\rho_i$ can vary substantially across layers depending on the underlying graph structure. In Erdős–Rényi graphs (ERDdS & R&wi, 1959), $\rho_i$ tends to concentrate around a global average value, yielding relatively uniform connectivity along the path. In contrast, stochastic block models(SBM graph (Holland et al., 1983)) exhibit much greater variability in $\rho_i$ due to community structure: layers aligned with intra-community regions have high density, whereas inter-community transitions form low-density bottlenecks, especially when paths traverse hub or boundary nodes.

Therefore, we explicitly control the proportion of samples used in our experiments to avoid underfitting caused by insufficient training data. We enumerate all subgraphs and shortest paths to construct datasets for intrinsic and extrinsic reasoning, respectively. Among all samples, 90% are used for training and the remaining 10% are reserved for evaluation. To prevent information leakage between training and testing, we further remove from the test set any shortest path whose node sequence is a subsequence of a training path. During training, we use a train ratio to regulate the fraction of available training samples, allowing us to systematically study the effect of data scale on model performance.

## 4. Hallucination in Intrinsic Reasoning

Reasoning over paths with given contextual constraints has been widely studied as a way to probe the reasoning capabilities of Transformer models (Wang et al., 2024; Saparov et al.; Sanford et al., 2024; Cohen et al.). Following this, we focus on analyzing what occurs during the training phase.

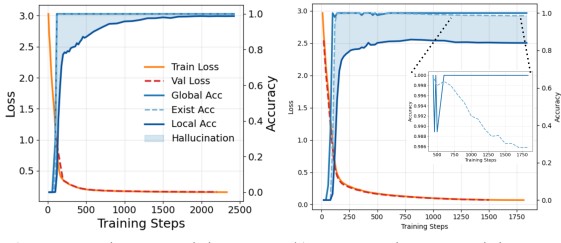

a) Accuracy in 20% training set    b) Accuracy in 0.1% training set

*Figure 2.* Training evolution in intrinsic reasoning. (a) Accuracy trajectories under a 20% training set. The persistent gap between global and local accuracy indicates the emergence of hallucinations. (b) Accuracy trajectories under a 0.1% training set. Hallucinations persist and cannot be resolved by simply increasing the number of training steps. The zoomed-out view further shows that, at later stages of training, the Exist Acc gradually deteriorates and Global Acc is not stable.

**Experiment Settings** First, we construct the underlying graph using an Erdős–Rényi (ER) graph (ERDdS & R&wi, 1959) with $N = 10$ nodes and edge probability $p = 0.4$. From this underlying graph, we enumerate 225,808 subgraphs, yielding a total of 3,982,697 training samples. Since all subgraphs are derived from the same underlying graph, we first evaluate whether the model can utilize edges that exist in the underlying graph, regardless of the imposed conditions. We denote this metric as $\text{ACC}_{\text{Exist}}$, which measures whether the predicted paths are valid with respect to the underlying graph structure. To further assess the model's understanding of the global graph structure, we define $\text{ACC}_{\text{Global}}$, which evaluates generalization over all reachable source–target node pairs. In this setting, each query is specified only by the source and target nodes as $q = (\text{S}, s, \text{T}, t, \text{PATH})$. Finally, to evaluate contextual understanding, we use $\text{ACC}_{\text{Local}}$, which measures whether the predicted paths strictly satisfy the constraints imposed by the given subgraph context. Formal definitions are provided in Section A, and the evolution of these metrics is shown in Figure 2a). We train from scratch on Llama-2 architecture, the full experiment setting is listed in the Appendix Section C.

**Observation from Training Dynamics** The results suggest that the Transformer eventually learns to perform path searching over the sampled subgraphs, consistent with prior studies (Wang et al., 2024). However, we observe that the model acquires a global view of the reasoning space before mastering local, condition-faithful reasoning. In the early stages of training, the Transformer often produces predictions that violate the given conditions, yet the predicted edges already exist in the underlying graph. Later, the Transformers start to learn reasoning paths with given conditions. This indicates that the Transformer first develops a global structural understanding of the graph, and only later learns to perform localized reasoning that requires sat-

isfying specific conditions. During this underfitting in the early stage, the model frequently reuses previously learned reasoning paths across different conditional queries, regardless of whether those paths are consistent with the given conditions. We refer to this phenomenon as **Path Reuse**.

We also observe that such hallucinations persist when training relies on only a limited number of samples, as shown in Figure 2b). Although both the training and validation losses continue to decrease, performance on the test set no longer improves, while the Global and Exist accuracies remain high. As discussed in Section 3.2, when the underlying graph is dense and contains hundreds of nodes with near-fully connectivity, the number of subgraphs grows rapidly, making it substantially harder for Transformers to avoid hallucinations if only sampling a small number of training data. This behavior reflects the difficulty of Transformers in performing systematic search (Saparov et al.). Concretely, we find that Exist accuracy gradually declines as training progresses, while Global accuracy becomes unstable in later stages, indicating that the Transformer begins to overfit to a subset of specific paths. We analyze this overfitting stage in detail in Section 5.

> **Takeaway 4.1**
>
> Path reuse hallucinations in intrinsic reasoning arise when Transformer underfitting, typically in early training or with limited data, where the model relies on edges from the underlying graph rather than the given context for reasoning.

# 5. Hallucination in Extrinsic Reasoning

In Section 4, we show that learning the underlying graph precedes the learning of local contextual reasoning. This indicates that the underlying graph establishes the foundational search capability of Transformers, suggesting that extrinsic reasoning constitutes their basic reasoning ability.

To further investigate how Transformers explore paths within the underlying graph, we design a set of experiments to analyze how graph structure is internalized during training Section 5.1 and to identify the emergence of path compression that leads to hallucinations in Section 5.2.

## 5.1. Path Compression in Extrinsic Reasoning

**Experiment Setting**  To investigate how Transformers memorize underlying graph from the training corpus in sequence data, we train a 6-layer LLaMA-2–style architecture from scratch. Following the guidance of Section 3.2, we construct the training set using the maximum amount of data available in the synthetic search space, ensuring full exposure to all possible path structures.

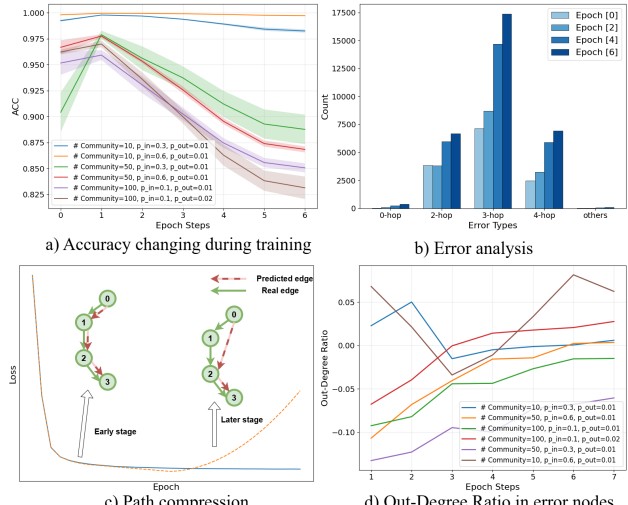

*Figure 3.* Path Compression. (a) Accuracy degradation during training across graph settings. (b) Errors are biased toward k-hop neighbors. (c) Path-compression hallucinations via implicit k-hop edge creation. (d) Accumulation of high–out-degree nodes in erroneous predictions.

We generate the underlying graphs using the Stochastic Block Model (SBM), with 1,000 nodes and varying numbers of communities, as well as different intra- and inter-community connection probabilities $p_{in}$ and $p_{out}$. For each underlying graph, we allocate 90% of the paths in the search space for training and reserve the remaining 10% for evaluation. Since the total number of training samples varies with graph properties, we measure training progress in terms of epoch steps, where each step corresponds to 7 full pass over the entire training set. Accuracy is evaluated under a strict criterion: a predicted path is considered correct only if all predicted edges exist in the underlying graph.

**Observation from Training Dynamics**  The results are summarized in Figure 3a). Each model was run three times; detailed results are provided in Appendix Table 3. Accuracy trajectories indicate that Transformers first achieve near-perfect accuracy across a wide range of graph configurations, followed by a steady degradation as training continues. Although different graph structures induce distinct data distributions that affect overall performance levels, this degradation consistently appears across settings, differing primarily in magnitude. For instance, when the graph contains fewer communities (e.g., 10) with a high intra-community probability ($p_{in} = 0.6$), accuracy decreases slightly from 1.0 to 0.98. In contrast, as the number of communities increases (e.g., 50), the model becomes less able to maintain high performance, with accuracy dropping from 0.97 to 0.85. A detailed analysis of how graph properties influence model performance is provided in Section E.

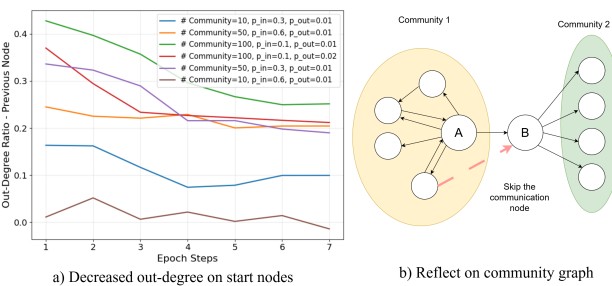

a) Decreased out-degree on start nodes

b) Reflect on community graph

*Figure 4.* Error Understanding. (a) Prediction errors decrease as the out-degree of the start nodes increases. (b) Path compression reveals a tendency to bypass bridge nodes, directly jumping across communities.

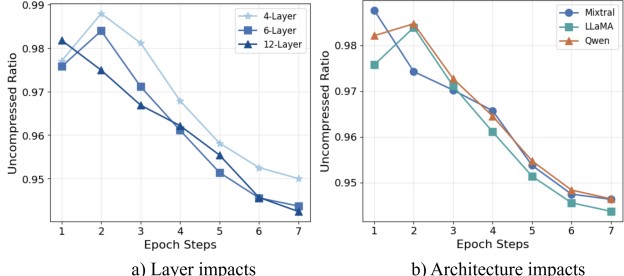

a) Layer impacts

b) Architecture impacts

*Figure 5.* Uncompressed ratio under different model settings. (a) Increasing the number of layers does not prevent the decline of the uncompressed ratio. (b) The uncompressed ratio consistently declines across different next-token prediction frameworks.

Further, we analyze the behavior of the model when its accuracy begins to decline by examining the properties of incorrectly predicted nodes across training epochs. Using a graph with 1000 communities and connection probabilities $p_{in} = 0.1$ and $p_{out} = 0.01$ as a representative example, we analyze error patterns throughout training, as shown in Figure 3b. Specifically, we categorize errors by the hop distance between the predicted node and the ground-truth target. A 0-hop error corresponds to repeating the previous node, while the "others" category includes predictions that cannot reach the target node or whose path length exceeds five hops. Among all error types, predicting 3-hop neighbors constitutes the most frequent failure mode. Moreover, the proportion of this error increases as training progresses. These results suggest that transformers increasingly favor predicting $k$-hop neighbors during training. As suggested in Figure 3c, in the overfitting stage, the transformers will easily take the $k$-hop nodes, choosing the higher out-degree nodes to be the next token, which is defined as **Path Compression**.

> **Takeaway 5.1**
>
> Path compression hallucinations in extrinsic reasoning arise when Transformer overfitting, typically in later training stages, where the model compresses multi-hop paths into k-hop neighbor connections.

### 5.2. Mechanism Exploring

**Factor Analysis**   Next, we investigate why Transformers compress paths by effectively introducing new edges. Specifically, we analyze this behavior through node out-degree statistics. We first apply z-score normalization to the out-degrees of all nodes in the underlying graph. We then collect the nodes involved in non-existent edges predicted by the model and compute the mean normalized out-degree of these nodes at each training epoch. The resulting out-degree ratios for the predicted nodes and the preceding nodes are visualized in Figure 3d) and Figure 4a), respectively.

The normalized out-degree of predicted nodes increases over training, while that of the previous nodes decreases. For example, in a graph with 1,000 communities and connection probabilities $p_{in} = 0.1$ and $p_{out} = 0.02$ (red curve), accuracy drops from 0.95 to 0.80 as training progresses. During the same period, the mean normalized out-degree of incorrectly predicted nodes rises from $-0.06$ to $0.025$, whereas that of the preceding nodes decreases from $0.35$ to $0.25$. These results indicate that path compression predominantly occurs when reasoning transitions skip low–out-degree nodes and directly jump to high–out-degree nodes, effectively bypassing community-level bridge nodes, as illustrated in Figure 4b).

Consequently, underlying graphs with many communities are more susceptible to path compression. As shown in Section 5.1 and Figure 3a), graphs with fewer communities exhibit significantly less path compression and maintain higher accuracy; for instance, graphs with only 10 communities are much more robust to this effect. We note that community structure is jointly determined by $p_{in}$ and $p_{out}$, and these factors also influence the severity of path compression. Additional results analyzing the impact of graph properties on path compression are provided in the Appendix E.

**Layer and Architecture Impacts**   We first investigate the effect of model depth, which is regard as a critical factor in LLMs reasoning (Chen et al., 2025). To quantify the extent to which reasoning paths remain uncompressed, we introduce the Uncompressed Ratio $R$, defined as $R = \frac{\text{predicted path length}}{\text{ground-truth path length}}$. A higher value of $R$ indicates that fewer reasoning paths are compressed by the model. Our results in Figure 5(a) show that deeper layers do not help decoder-only Transformers recover from path compression. In contrast, shallower layers mitigate compression effects that would otherwise accumulate as training converges.

We further compare several model architectures, including LLaMA-2, Qwen-2 (which adopts a different positional embedding scheme from LLaMA), and Mixtral (which in-

---

**Example**

**Path in underlying graph**: PATH 33 381 76 62 END
**Build the Relations**: Kimberly King (33) collaborated with Marlene Beck (381); Marlene Beck friend of Scott Cox (76); Scott Cox supervised by Gary Murphy (62).
**Generated story**: Kimberly King, a rising star, always valued Marlene Beck's sharp insights, their collaborations sparking innovation. Marlene, in turn, found a kindred spirit in Scott Cox, their friendship a bedrock of mutual support. Scott, navigating his career, often leaned on Gary Murphy, his patient supervisor. Gary, observing Scott's dedication, saw a reflection of his own early days. Their intertwined paths, fueled by collaboration, friendship, and guidance, painted a quiet portrait of professional and personal growth.

---

*Figure 6.* Example of changing a reasoning path from the underlying graph to a story.

corporates a mixture-of-experts design). While the overall trends are consistent across architectures, subtle differences emerge in their training dynamics. Specifically, Mixtral achieves its highest uncompressed ratio in the first training epoch, whereas LLaMA-2 and Qwen-2 peak in the second epoch. Nevertheless, all models exhibit a progressively increasing degree of path compression as training proceeds.

**Path Compression Hypothesis** We attribute the path-skipping phenomenon to models' sequence co-occurrence rather than strict topological adjacency. To learn path searching, the model must implicitly learn the underlying graph structure from the training sequences. However, because the model aggregates information across a context window, the model's learned transition probabilities do not strictly bound themselves to a 1-step adjacency matrix. Instead, they effectively approximate a convex combination of multi-step graph transitions. We hypothesize that skipping occurs when this learned multi-hop context conflates with the 1-step prediction space. Specifically, consider a subsequent node $u$ that is $k$ hops away from the current node $v$. Because $u$ frequently co-occurs with $v$ in the training paths, the model assigns a significant weight to this $k$-step relationship. If the aggregated multi-step probability mass for $u$ outweighs the sparse 1-hop transition probability to the true intermediate neighbor $m$, the statistical co-occurrence effectively overrides the discrete topological constraints. Consequently, the model hallucinates a shortcut directly to $u$—predicting a non-existent edge because its representation space correctly captured the multi-hop correlation but failed to respect the hard sparsity of the graph (see Appendix B for an analytical model and proofs).

---

**Takeaway 5.2**

Path compression hallucinations occur more frequently across communities by jumping over bridge nodes. Their occurrence reflects a bias induced by the training data, rather than an effect of different Transformer architectures.

---

# 6. Application Impact and Discussion

We further examine how hallucinations modeled by underlying graphs inform our understanding of the mechanisms

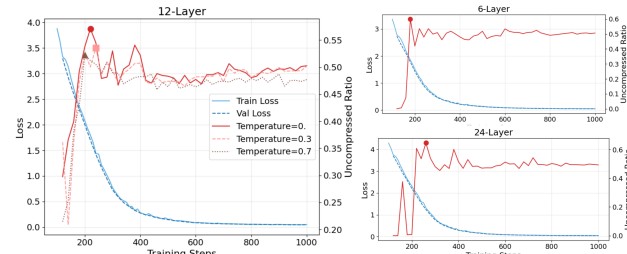

*Figure 7.* Training on language sequence. The highest uncompressed ratio is in the early stage.

underlying LLM behavior. Specifically, we analyze how hallucinations emerge during language-sequence pretraining in Section 6.1, how fine-tuning methods mitigate them in Section 6.2, and how these mechanisms relate to well-known phenomena in large language models in Section 6.3.

## 6.1. The Effects of Path Compression in Pre-training Language Sequence

To analyze the impact of path compression on language models in real-world settings, we convert synthetic graphs into natural-language stories. We construct a 500-node stochastic block model with 25 communities ($p_{in} = 0.1, p_{out} = 0.01$), where each node represents a person entity. We define 40 types of interpersonal relations (e.g., friends, peers, coworkers), which are assigned to edges in the underlying graph. For each sampled path, we instantiate the corresponding sequence of entity relations. Finally, we use Gemini-2.5-lite to generate stories from these relational paths, producing natural-language training samples. The input prompt is:"Giving the people relation path:[Relations]. Please write an 80-word story of them based on the path.". Figure 6 illustrates an example of this translation process.

We train Qwen3-style Transformers from scratch to study how they memorize underlying knowledge graphs from language sequences, evaluating models with depths ranging from 6 to 24 layers. To evaluate the reliable memorized knowledge, we prompt the model with a person's name as the starting entity and let it generate a related narrative. We assess whether the generated person names follow the correct order implied by the underlying relational paths. In addition, we evaluate generation behavior under different decoding temperatures ($T = 0, 0.3, 0.7$). The results are shown in Figure 7. We highlight the maximum uncompressed path ratio with special markers.

The results show that the maximum uncompressed path ratio is achieved at an early stage of training, even when both training and validation losses have already reached relatively low values. After loss convergence, the uncompressed path ratio stabilizes but no longer reaches the earlier peak performance. This behavior is consistent across models with different numbers of layers. In real-world corpora, auxiliary

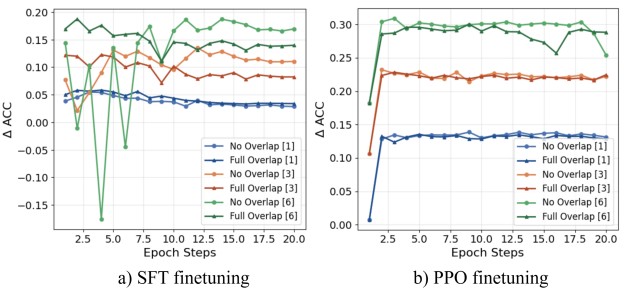

a) SFT finetuning      b) PPO finetuning

*Figure 8.* Accuracy improvement ($\Delta$ACC) during finetuning with SFT and PPO. The original Acc are 0.8527, 0.753, 0.6787 at epoch [1][3][6]. (a) SFT finetuning shows unstable recovery behavior, especially at later pretraining stages. (b) PPO finetuning consistently achieves higher and more stable accuracy gains across different hop distances.

tokens introduce additional noise and interference. Thus, while the overall loss decreases, path compression remains observable at the knowledge graph level. However, this loss reduction does not imply faithful knowledge understanding; instead, the Transformers exhibit path-compression hallucinations. Meanwhile, we include an analysis of real-world language data distributions in Appendix D.

## 6.2. Finetuning Recovery

We next investigate whether path compression hallucinations can be alleviated through different finetuning strategies. Experiments are conducted on graphs with 1000 nodes and 10 communities, with $p_{in} = 0.1$ and $p_{out} = 0.01$. During pretraining, the model is exposed to $50\%$ of all valid paths. In the finetuning stage, only $20\%$ of paths are available, drawn either from the same data distribution as pretraining (denoted as *full overlap*) or from a disjoint distribution (denoted as *no overlap*).

As shown in Figure 8, we compare SFT and PPO finetuning at three pretraining checkpoints (epochs 1, 3, and 6). Overall, PPO achieves more effective recovery than SFT, with the maximum accuracy improvement reaching approximately 0.3, whereas SFT peaks around 0.2. Notably, SFT exhibits clear collapse behavior when applied to models that have already been strongly affected by path compression, particularly at the later pretraining stage (epoch 6). The choice of finetuning data further amplifies this effect. While overlapping data can stabilize SFT in early stages, finetuning on out-of-domain data often leads to severe performance degradation (Zhang et al., 2026). In contrast, at later stages, previously unseen data can provide stronger regularization benefits than overlapping data. PPO shows substantially better generalization (Kirk et al., 2024). Since its optimization objective directly rewards the existence of valid edges, its performance is largely insensitive to the data overlap ratio, resulting in stable and consistent improvements across all settings. Overall, both fine-tuning and PPO are effective in

recovering from path-compression hallucinations; however, PPO yields more robust and stable improvements.

### 6.3. Connections to Related Phenomena

Based on the underlying graph assumption, we can provide unified insights into several previously observed behaviors of large language models.

**Reversal Curse** The reversal curse refers to the phenomenon that when a model is trained on relations of the form "$a$ is $b$", it often fails to generalize to the inverse relation "$b$ is $a$" (Berglund et al.). From an underlying graph perspective, such relational facts correspond to directed transitions between nodes. Although the full relation implicitly forms a cycle between $a$ and $b$, training data typically supervises only one direction (e.g. $a \rightarrow b \rightarrow c \rightarrow a$). At early stages of training, the model tends to represent the relation through multi-step paths that traverse intermediate nodes along this cycle. However, as training progresses, path compression occurs: the multi-step reasoning path is collapsed into a single directed edge aligned with the observed supervision (e.g. $a \rightarrow b \rightarrow a$). As a result, the inverse direction remains weakly represented as it happens at later stages, requiring substantially more training signal to emerge. This explains why learning "$b$ is $a$" is empirically harder and more sample-inefficient.

**Reasoning Distribution Bias** The underlying graph framework explicitly models reasoning as a structured set of transitions (Kalai et al., 2025). Under this view, the reasoning behavior of decoder-only Transformers is closely coupled with the degree distribution of the underlying graph. Nodes with higher in- or out-degrees receive disproportionately more supervision during next-token prediction, while low-degree or peripheral nodes are underrepresented. Since real-world language data induces a highly non-uniform underlying graph, this leads to systematic reasoning biases. In particular, transitions involving low-degree nodes or rare relational patterns are more prone to omission or hallucination, as they are weakly encoded during training. This observation suggests that limitations in LLM reasoning are not solely due to model capacity, but also arise from structural biases inherited from the underlying reasoning graph.

## 7. Conclusion

We analyze reasoning hallucinations in decoder-only Transformers from an underlying graph perspective, modeling next-token prediction as a graph search over implicit relational structures. This formulation unifies intrinsic reasoning, which is reasoning under contextual constraints, and extrinsic reasoning, which relies on memorized underlying knowledge. We identify two core mechanisms behind

hallucinations: Path Reuse, where memorized paths dominate early training, and Path Compression, where frequent multi-step paths collapse into shortcuts later in training. Our results highlight fundamental limitations of next-token prediction for faithful multi-step reasoning and motivate future work on structurally grounded training objectives.

## Impact Statement

This work provides a mechanistic analysis of reasoning hallucinations in decoder-only Transformers by modeling next-token prediction as a graph search over implicit relational structures. Our findings can help diagnose unfaithful reasoning and inform the design of more reliable evaluation and training methods for large language models. Potential risks include misuse of these insights to elicit convincing but unsupported reasoning, or overestimating their ability to prevent hallucinations. We emphasize that our contribution is explanatory rather than a complete solution, and should be combined with existing practices in real-world deployments.

## Acknowledgements

Xinnan Dai, Shenglai Zeng, Kai Guo and Jiliang Tang are supported by the National Science Foundation (NSF) under grant numbers CNS2321416, IIS2212032, IIS2212144, IIS 2504089, DUE2234015, CNS2246050, DRL2405483 and IOS2035472, the Michigan Department of Agriculture and Rural Development, US Dept of Commerce, Gates Foundation, Amazon Faculty Award, Meta, NVIDIA, Microsoft and SNAP.

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

## A. Evaluation Metrics

**Notation.** Let $G = (V, E)$ denote the backbone graph with $|V| = N$. Each conditional reasoning query is defined as

$$q = (s, t, \mathcal{C}), \tag{1}$$

where $s, t \in V$ are the source and target nodes, and $\mathcal{C}$ denotes a set of conditional constraints. Given a query $q$, the model predicts a path

$$\hat{P}_{s \to t} = (v_0 = s, v_1, \ldots, v_k = t). \tag{2}$$

We define an indicator function for path validity in the backbone graph:

$$\mathbb{I}_{\text{path}}(\hat{P}, G) = \begin{cases} 1, & \text{if } (v_i, v_{i+1}) \in E, \ \forall i, \\ 0, & \text{otherwise.} \end{cases} \tag{3}$$

Similarly, we define an indicator for condition satisfaction:

$$\mathbb{I}_{\text{cond}}(\hat{P}, \mathcal{C}) = \begin{cases} 1, & \text{if } \hat{P} \text{ satisfies } \mathcal{C}, \\ 0, & \text{otherwise.} \end{cases} \tag{4}$$

**Local Accuracy.** Local accuracy measures whether the predicted path both exists in the backbone graph and strictly satisfies the given conditions:

$$\text{Acc}_{\text{Local}} = \frac{1}{|\mathcal{Q}|} \sum_{q \in \mathcal{Q}} \mathbb{I}_{\text{path}}(\hat{P}_q, G) \cdot \mathbb{I}_{\text{cond}}(\hat{P}_q, \mathcal{C}_q), \tag{5}$$

where $\mathcal{Q}$ denotes the set of evaluated queries.

**Exist Accuracy.** Exist accuracy relaxes the conditional constraint and only requires the predicted path to exist in the backbone graph:

$$\text{Acc}_{\text{Exist}} = \frac{1}{|\mathcal{Q}|} \sum_{q \in \mathcal{Q}} \mathbb{I}_{\text{path}}(\hat{P}_q, G). \tag{6}$$

**Global Accuracy.** Global accuracy evaluates reasoning performance over all reachable node pairs in the backbone graph, independent of the sampled conditions:

$$\text{Acc}_{\text{Global}} = \frac{1}{|\mathcal{V}_{\text{reach}}|} \sum_{(u,v) \in \mathcal{V}_{\text{reach}}} \mathbb{I}_{\text{path}}(\hat{P}_{u \to v}, G), \tag{7}$$

where

$$\mathcal{V}_{\text{reach}} = \{(u, v) \mid u \neq v, \ \exists \text{ a path from } u \text{ to } v \text{ in } G\}. \tag{8}$$

## B. A Simplified Model of Path Compression Phenomenon

In this section, we provide a simplified mathematical model to explain the skipping phenomenon observed during inference. We first demonstrate the optimal substructure property of shortest paths, which justifies modeling the generative process as a sequence of localized transitions.

**Proposition B.1** (Substructure Property of Shortest Paths). *Let $\pi^* = (v_1, \ldots, v_L)$ be a shortest path in a graph. For any intermediate node $v_k$ where $1 \leq k < L$, the suffix $(v_k, \ldots, v_L)$ remains a strictly optimal shortest path from $v_k$ to $v_L$.*

*Proof.* Assume, for the sake of contradiction, that there exists an alternative path $\pi'$ from $v_k$ to $v_L$ such that the length of $\pi'$ is strictly less than the length of the suffix $(v_k, \ldots, v_L)$. By concatenating the prefix $(v_1, \ldots, v_k)$ with $\pi'$, we would construct a valid path from $v_1$ to $v_L$ whose total length is strictly shorter than $\pi^*$. This contradicts the premise that $\pi^*$ is a shortest path. Therefore, the optimal suffix is independent of the prefix, establishing that the optimal next hop depends solely on the current node $v_k$ and the target node. $\square$

Based on this optimal substructure, we can formulate the autoregressive generation of the model, parameterized by $\theta$, as predicting the next optimal step $P_\theta(v_{k+1} \mid v_k)$, effectively discarding the sequence history.

Let $T$ denote the true one-step random walk transition matrix of the underlying graph, where $(T)_{xy} = \frac{1}{\text{outdeg}(x)}$ if an edge $x \to y$ exists, and $0$ otherwise. Because Transformers utilize self-attention to capture multi-hop co-occurrences across a context window $K$, we assume the model's learned transition probabilities implicitly approximate a convex combination of multi-step random walks.

**Assumption B.2** (Mixture of Transition Matrices). Let $P_\theta$ be the transition probability predicted by the model. We assume $P_\theta(y \mid x)$ takes the form of a weighted mixture of $i$-step transitions:

$$P_\theta(y \mid x) = \sum_{i=1}^{K} \lambda_i (T^i)_{xy}$$

where $\lambda_i \geq 0$ are learned weights representing the influence of the $i$-hop neighborhood.

**Proposition B.3** (Condition for Hallucinating Shortcuts). *Assume a context window $K = 2$. Let $v$, $m$, and $u$ be nodes such that there is a 2-hop path $v \to m \to u$, but no direct edge $v \to u$. Furthermore, assume there is no 2-hop path from $v$ to $m$ (i.e., $(T^2)_{vm} = 0$). If the learned weights $\lambda_i$ satisfy:*

$$\frac{\lambda_2}{\lambda_1} > \frac{1}{\sum_{m' \in \mathcal{M}} \frac{1}{\text{outdeg}(m')}}$$

*where $\mathcal{M}$ is the set of all intermediate nodes connecting $v$ to $u$ (i.e., $v \to m' \to u$), then the model will predict the skipped node with higher probability than the true adjacent node: $P_\theta(u \mid v) > P_\theta(m \mid v)$.*

*Proof.* By Assumption B.2 with $K = 2$, and the assumption that $(T^2)_{vm} = 0$, the model's predicted probability for the true 1-hop neighbor $m$ is:

$$P_\theta(m \mid v) = \lambda_1(T)_{vm} + \lambda_2(T^2)_{vm} = \lambda_1(T)_{vm} = \frac{\lambda_1}{\text{outdeg}(v)}.$$

Similarly, since there is no direct edge $v \to u$, the model's predicted probability for the 2-hop target $u$ is:

$$P_\theta(u \mid v) = \lambda_1(T)_{vu} + \lambda_2(T^2)_{vu} = \lambda_2(T^2)_{vu} = \lambda_2 \sum_{m' \in \mathcal{M}} T_{vm'}T_{m'u} = \frac{\lambda_2}{\text{outdeg}(v)} \sum_{m' \in \mathcal{M}} \frac{1}{\text{outdeg}(m')}.$$

Therefore, when

$$\frac{\lambda_2}{\lambda_1} > \frac{1}{\sum_{m' \in \mathcal{M}} \frac{1}{\text{outdeg}(m')}},$$

we have $P_\theta(u \mid v) > P_\theta(m \mid v)$. $\square$

**Proposition B.4** (Generalized Condition for Path Compression Hallucination). *Let the context window be $K \geq 2$. Consider a source node $v$ and a target node $u$ such that the shortest path distance from $v$ to $u$ is $d$, where $2 \leq d \leq K$. Let $m$ be a true 1-hop neighbor of $v$ on the shortest path.*

*Assume there are no cycles of length $\leq K$ returning to $m$ from $v$ (i.e., $(T^i)_{vm} = 0$ for $2 \leq i \leq K$). Let $\Pi_d(v, u)$ denote the set of all valid $d$-hop paths from $v$ to $u$. For any path $\pi \in \Pi_d(v, u)$, let $\mathcal{I}(\pi)$ be the set of its intermediate (interior) nodes.*

*If the learned mixture coefficients satisfy*

$$\frac{\lambda_d}{\lambda_1} > \frac{1}{\sum_{\pi \in \Pi_d(v,u)} \prod_{x \in \mathcal{I}(\pi)} \frac{1}{\text{outdeg}(x)}},$$

*then the model will predict the skipped node $u$ with higher probability than the true adjacent node $m$: $P_\theta(u \mid v) > P_\theta(m \mid v)$.*

*Proof.* By Assumption B.2 and the assumption that $(T^i)_{vm} = 0$ for $2 \leq i \leq K$, the model's predicted probability for the true 1-hop neighbor $m$ is:

$$P_\theta(m \mid v) = \sum_{i=1}^{K} \lambda_i (T^i)_{vm} = \lambda_1(T)_{vm} = \frac{\lambda_1}{\text{outdeg}(v)}.$$

| Hyperparameter | Value |
|---|---|
| Hidden size ($d_{\text{model}}$) | 512 |
| Sequence length (block size) | 128 |
| Learning rate | $5 \times 10^{-4}$ |
| Optimizer | AdamW |
| Weight decay | 0.1 |
| Learning rate scheduler | Cosine |
| Warmup ratio | 0.03 |
| Batch size (per device) | 2048 |
| Gradient accumulation steps | 1 |
| Evaluation strategy | Step-based |
| Checkpoint saving interval | One epoch |
| Precision | bfloat16 (bf16) |
| Gradient checkpointing | Enabled |
| Dynamic saving | Disabled |

Table 1. Training hyperparameters used in all experiments unless otherwise specified.

For the target node $u$, since the shortest path distance is $d$, we know that $(T^i)_{vu} = 0$ for all $1 \le i < d$. Thus we have

$$P_\theta(u \mid v) = \sum_{i=d}^{K} \lambda_i (T^i)_{vu} \ge \lambda_d (T^d)_{vu} = \frac{\lambda_d}{\text{outdeg}(v)} \sum_{\pi \in \Pi_d(v,u)} \prod_{x \in \mathcal{I}(\pi)} \frac{1}{\text{outdeg}(x)}$$

Therefore, when

$$\frac{\lambda_d}{\lambda_1} > \frac{1}{\sum_{\pi \in \Pi_d(v,u)} \prod_{x \in \mathcal{I}(\pi)} \frac{1}{\text{outdeg}(x)}},$$

we have

$$P_\theta(u \mid v) \ge \frac{\lambda_d}{\text{outdeg}(v)} \sum_{\pi \in \Pi_d(v,u)} \prod_{x \in \mathcal{I}(\pi)} \frac{1}{\text{outdeg}(x)} > \frac{\lambda_1}{\text{outdeg}(v)} = P_\theta(m \mid v),$$

as desired. □

## C. Experiment setting

Here we list our parameter settings in the experiments in Tables 1 and 2. All of the training is on one H200. The test is on 8 A6000 as we evaluate the Transformers' behaviors at every stage. For each graph setting, we run three different random seeds and report the mean and standard deviation of the accuracy. In addition, we report the minimum runtime observed across the three seeds. The result details are shown in 3.

## D. Real-World Connections

Our assumption is that LLMs will set up the inner knowledge graph while learning the data from language sequences; therefore, we take the DocReD dataset (Yao et al., 2019), which has both documents and a manually extracted knowledge graph. We train a 4-layer Qwen3 model on the documents and evaluate the knowledge uncompression ratio following section 6.1. We select 500 high-frequency entity nodes (more than 4 documents include the entities) to evaluate the uncompression ratio, as shown in Figure 9.The results suggest that, during real-world language training, knowledge is well organized before the model begins to overfit. The uncompression ratio is significantly higher than in the overfitting stage.

Meanwhile, we find that real-world language training data can indeed be structured as community graphs. For the math and code datasets, we construct token-level graphs by filtering out low-frequency tokens (occurring fewer than 10 times). For the manually built knowledge graph from DocRED, we remove isolated nodes. We then apply the Leiden algorithm to identify community structures, as summarized in Table 4. The results demonstrate that language data can be clearly partitioned into multiple communities, revealing structured patterns as discussed in the paper.

| Parameter | LLaMA | Mixtral | Qwen2 |
|---|---|---|---|
| Hidden size ($d_{\mathrm{model}}$) | | 512 | |
| FFN hidden size | $2.8 \times d_{\mathrm{model}}$ | $2.8 \times d_{\mathrm{model}}$ | $2.8 \times d_{\mathrm{model}}$ |
| Number of layers | | 6 | |
| Attention heads | | 16 | |
| Key-value heads (GQA) | | 8 | |
| Head dimension | | 64 | |
| Max position embeddings | | 128 | |
| RoPE $\theta$ | | 10,000 | |
| RMSNorm $\epsilon$ | $1 \times 10^{-5}$ | $1 \times 10^{-5}$ | $1 \times 10^{-6}$ |
| Activation function | SiLU | SiLU | SiLU |
| KV cache during training | Disabled | Disabled | Disabled |
| **MoE-specific** | – | | – |
| Number of experts | – | 8 | – |
| Experts per token | – | 2 | – |
| Routing type | – | Top-$k$ | – |
| Router auxiliary loss | – | $1 \times 10^{-2}$ | – |
| Router jitter | – | 0.0 | – |

*Table 2.* Backbone model configurations for different Transformer architectures.

| #community$_{-}p_{in-}p_{out}$ | epoch-1 | epoch-7 | epoch-49 | time (ms) |
|---|---|---|---|---|
| 10_0.3_0.01 | $0.992 \pm 0.00$ | $0.998 \pm 0.00$ | $0.983 \pm 0.00$ | 1197.10 |
| 10_0.6_0.01 | $0.998 \pm 0.00$ | $0.999 \pm 0.00$ | $0.997 \pm 0.00$ | 1805.54 |
| 50_0.3_0.01 | $0.904 \pm 0.02$ | $0.979 \pm 0.04$ | $0.888 \pm 0.14$ | 324.62 |
| 50_0.6_0.01 | $0.967 \pm 0.07$ | $0.978 \pm 0.02$ | $0.898 \pm 0.03$ | 375.81 |
| 100_0.1_0.01 | $0.952 \pm 0.01$ | $0.959 \pm 0.01$ | $0.851 \pm 0.00$ | 199.79 |
| 100_0.1_0.02 | $0.962 \pm 0.002$ | $0.970 \pm 0.003$ | $0.831 \pm 0.011$ | 393.95 |

*Table 3.* Performance under different graph structures

# E. Graph Property Effects

As discussed in Section 5.2, path compression is closely tied to the structure of the underlying graph. Here, we further investigate path compression under different parameter settings of stochastic block model (SBM) graphs. All experiments are conducted with fully trained Transformer models, and the results are shown in Figures 10 to 12. The k-ratio is defined as the ratio of the number of communities to the number of nodes, i.e., #community/#nodes. We additionally examine whether Transformers benefit from increased training across these settings.

Overall, graphs with clearer community structure—characterized by smaller k-ratio, lower $p_{\mathrm{out}}$, and higher $p_{\mathrm{in}}$—exhibit greater training stability and are less prone to path-compression–induced hallucinations.

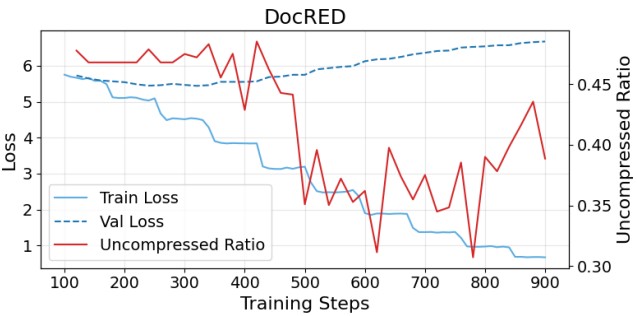

*Figure 9.* Path compression in knowledge graph data.

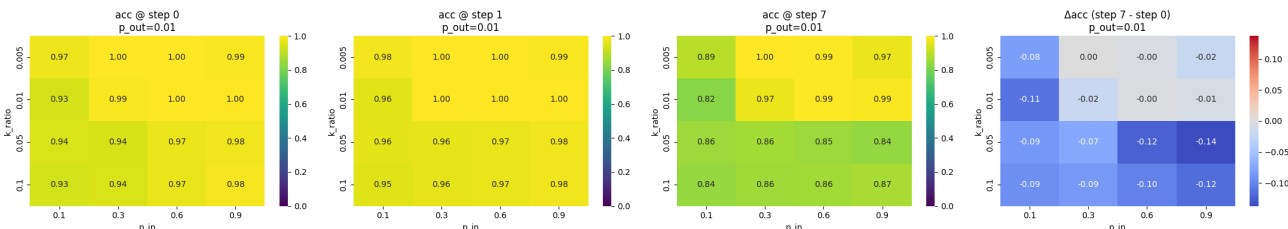

*Figure 10.* Underlying graphs with 0.01 $p_{out}$

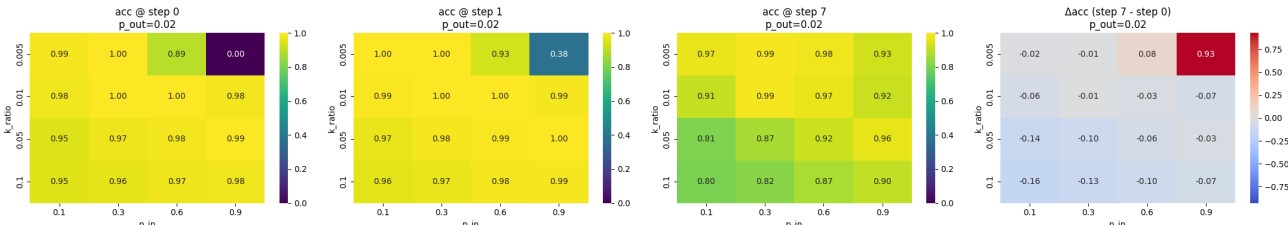

*Figure 11.* Underlying graphs with 0.02 $p_{out}$

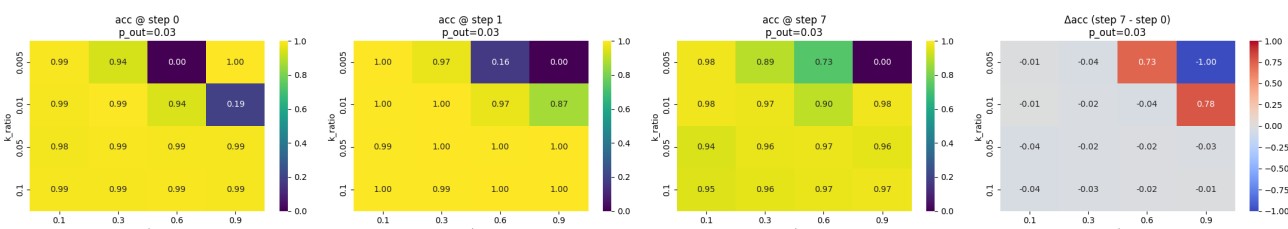

*Figure 12.* Underlying graphs with 0.03 $p_{out}$

| Dataset | #Node | #Edge | #Community | Internal degree | External degree |
|---|---|---|---|---|---|
| GSM8K (Math) | 1406 | 5084 | 71 | 0.383 | 0.002 |
| APPS (Code) | 12454 | 112184 | 136 | 0.389 | 0.001 |
| DocRED (KG) | 20221 | 38180 | 770 | 0.431 | $1 \times 10^{-5}$ |

*Table 4.* Real world data analysis

