# OpenReview forum: "When Do Hallucinations Arise? A Graph Perspective on the Evolution of Path Reuse and Path Compression"
_ICML.cc/2026/Conference — ICML 2026 regular_

### Official Review · Reviewer_sY6f · 2026-03-10

**Soundness:** 3
**Presentation:** 3
**Significance:** 3
**Originality:** 3
**Overall Recommendation:** 4
**Confidence:** 3

**Summary:**

This paper investigates the mechanisms behind reasoning hallucinations in decoder-only large language models (LLMs) by modeling next-token prediction as a graph search process over an underlying graph.  The authors distinguish between intrinsic reasoning (contextual, constrained search over a sampled subgraph) and extrinsic reasoning (context-free queries relying on memorized graph structures). They propose that reasoning hallucinations stem from two primary phenomena: "Path Reuse" during early training, where memorized knowledge overrides contextual constraints, and "Path Compression" in later stages, where multi-step paths collapse into frequent shortcut edges. The study uses synthetic graphs and generated natural-language narratives to train Transformer models from scratch, ultimately connecting these mechanisms to known LLM behaviors. The paper provides insights into the dynamics of reasoning hallucinations and suggests that they may be an inherent aspect of LLM training.

**Compliance With Llm Reviewing Policy:**

Affirmed.

**Final Justification:**

Although the authors provided experiments trained on real knowledge graphs, there are still differences between structured graph data and real natural language documents. I agree with Reviewer YzNK that this paper needs to strengthen the discussion regarding synthetic data and real natural language data to improve the credibility of the proposed controlled environment. Therefore, I am lowering my score to Weak Accept.

**Key Questions For Authors:**

1. How do you justify the assumption that a deviation from the explicit graph path (Path Compression) strictly constitutes a "hallucination,"  especially if the model arrives at a factually correct final answer through learned parametric shortcuts? Does correct reasoning inherently require the output of the full sequence?

2. Are there plans to evaluate this framework on established, real-world Knowledge Graph completion benchmarks using pre-existing foundation models to verify if the synthetic findings translate to real-world applications?

3. Can you provide rigorous proof that the graph-simulated training data can be used to approximate the real-world natural language training data? Can the graph structural pattern also be found in the NL data?

**Limitations:**

More comprehensive theoretical analysis and stronger links to natural language data are needed to enhance the credibility of the findings.

**Strengths And Weaknesses:**

## Strengths

* Novel Theoretical Framework: The paper provides a cohesive, unified graph-based perspective to explain the internal search and reasoning behaviors of Transformers. Translating generation trajectories into topological paths is a creative way to mathematically ground the concept of hallucinations.


* Controlled Experimental Design: The use of synthetic structures, such as Erdős-Rényi and Stochastic Block Models, allows for rigorous isolation and observation of how network topology and community structures influence learning dynamics and path compression.

* Explanatory Power for Empirical Phenomena: The authors successfully draw compelling connections between their theoretical findings on graph structures and established LLM flaws, offering a logical structural explanation for issues like the reversal curse and reasoning distribution bias.

## Weakness

* Validity of the Output-as-Reasoning Assumption: The central premise assumes that the explicit output path directly corresponds to the LLM's internal reasoning process. However, LLM reasoning occurs within its parameter space, and output tokens are probabilistic mappings rather than strict step-by-step logical traces [1]. An LLM might learn complex, implicit patterns to infer valid relationships without outputting the exact predefined intermediate sequence. Consequently, the observed "Path Compression," where a model skips a bridge node to directly output a high-out-degree node, might not necessarily represent a detrimental hallucination in real-world scenarios. If the final derived answer is correct, the compressed path may simply reflect a valid, learned heuristic rather than a reasoning failure.

* Lack of Real-World Knowledge Graph (KG) Evaluation: The empirical validation relies entirely on simulated graph datasets. The study lacks experiments evaluating real, off-the-shelf LLMs on authentic Knowledge Graphs. In a real-world KG context, an LLM might directly and correctly predict the relationship between two entities without explicitly generating the topological path. Under the paper's current strict evaluation criteria—where a path is only correct if all predicted edges sequentially exist in the underlying graph —such direct, accurate answers would be erroneously penalized.

* Generalizability to Unstructured Data: The models tested (LLaMA-2, Mixtral, Qwen-2 architectures) are trained from scratch on highly structured data and relational paths that are translated into simplistic, formulated narratives. There is significant doubt regarding whether the observed training dynamics, specifically the distinct phases of Path Reuse (underfitting) and Path Compression (overfitting), will map accurately to foundation LLMs pretrained on massive, messy, and highly unstructured natural language corpora.

* Minor: There is an error in the core example provided in Figure 1c. The prompt text is stated as "Question: What is the relation from John to Linda?", but the corresponding "Ground Truth Reasoning" is listed as "David $\rightarrow$ John $\rightarrow$ Kim $\rightarrow$ Linda". Based on the reasoning chain provided, the question should be asking for the relation from David to Linda, not John.

[1] Kambhampati, Subbarao, et al. "Stop Anthropomorphizing Intermediate Tokens as Reasoning/Thinking Traces!." arXiv preprint arXiv:2504.09762 (2025).

Despite the concerns, I still believe the paper offers an interesting framework and a controlled experiment to effectively analyze hallucination issues in LLMs and their potential connection to training data. However, more comprehensive theoretical analysis and stronger links to natural language data are needed to enhance the credibility of the findings.

---

> ### Author Rebuttal · Authors · 2026-03-30
>
> > W1 & Q1: Validity of the Output-as-Reasoning Assumption; why explicit graph path to be "hallucination"
>
> A: Thank you for your valuable insights. We would like to clarify that our study provides a data distribution perspective to explain why LLM reasoning does not always align with underlying knowledge and does not strictly follow step-by-step logical traces. Based on our hypothesis, we observe that skipped reasoning stages are caused by high out-degree nodes, which represent a particular structural pattern in the underlying knowledge graph and correspond to a specific type of reasoning data distribution[2].
>
> One of our core assumptions is that the knowledge supporting reasoning steps within LLMs should remain consistent across different scenarios, and that the explicit outputs should accurately reflect how this knowledge is learned. Accordingly, we expect LLM outputs to align with their internally consistent knowledge. The validity of reasoning steps in our framework is evaluate based on whether they align with the underlying knowledge, rather than solely on whether the final answer is correct.  If the path compression happened, it is in conflict with the prior knowledge, causing Fact-Conflicting Hallucination, as defined in[3].
>
> Building on this assumption, our synthetic approach provides a white-box method for understanding the relationship between underlying knowledge distributions and observed model behavior, offering an explanation for phenomena such as “path compression.” As discussed in Section 5 of [1], a key challenge lies in the opacity of pretraining data and the difficulty of linking reasoning behavior to it. Our method helps address this issue by offering a more interpretable framework.
>
> Moreover, the reliability of reasoning chains is crucial for improving LLM capabilities, as demonstrated in [4][5]. We believe that our characterization of data distributions—particularly those involving high out-degree nodes—can contribute to better evaluation and verification of reasoning chain quality.
> >W2&W3&Q2&Q3: connection with real-world data/knowledge graph completion task
>
> A: Thanks for your valuable suggestions. We would like to add external experiments on the real-world knowledge graph. Our assumption is that LLMs will set up the inner knowledge graph while learn the data from language sequences; therefore, we take the DocReD dataset[6], which has both documents and a manually extracted knowledge graph. We train a 4-layer Qwen3 model on the documents and evaluate the knowledge uncompression ratio following section 6.1. We select 500 high-frequency entity nodes (more than 4 documents include the entities) to evaluate the uncompression ratio as shown in the table below.
>
> |epoch|20|100|1000|
> |-|-|-|-|
> |uncompression ratio|.467|.478|.402|
>
> The results suggested a similar curve to our synthetic graph, where it increases at the early stage, but decreases in the late stages.
>
> Meanwhile, we find that real-world language training data can indeed be structured as community graphs. For the math and code datasets, we construct token-level graphs by filtering out low-frequency tokens (occurring fewer than 10 times). For the manually built knowledge graph from DocRED, we remove isolated nodes.
>
> We then apply the Leiden algorithm to identify community structures, as summarized in the table below.
>
> |Dataset|#Node|#Edge|#Community|Interal degree|External degree|
> |-|-|-|-|-|-|
> |GSM8K (Math)|1406|5084|71|0.383|0.002|
> |APPS (Code)|12454|112184|136|0.389|0.001|
> |DocRED (KG)|20221|38180|770|0.431|1E-05|
>
> The results demonstrate that language data can be clearly partitioned into multiple communities, revealing structured patterns as discussed in the paper. Analyzing the distribution of real-world language data in this manner offers a promising direction for future research.
>
> However, our framework does not readily adapt to knowledge graph completion tasks. Such tasks are typically provided only with a graph and aim to infer potential edges, whereas our framework for large language models is based on training over sentences. Nonetheless, we hypothesize that a latent knowledge graph may exist within LLMs. Exploring this underlying structure is an intriguing direction for future work.
>
> We will add the experiments and discussions in the revision.
> >W4: Minor
>
>  A: Thank you for pointing this out. We appreciate the careful reading. We will modify this in the revision.
>
> [1]Stop Anthropomorphizing Intermediate Tokens as Reasoning/Thinking Traces!
>
> [2]Topology of Reasoning: Understanding Large Reasoning Models through Reasoning Graph Properties
>
> [3]Siren’s Song in the AI Ocean: A Survey on Hallucination in Large Language Models
>
> [4]s1: Simple test-time scaling
>
> [5]Deepseek-r1: Incentivizing reasoning capability in llms via reinforcement learning
>
> [6]DocRED: A large-scale document-level relation extraction dataset

---

> > ### Author Rebuttal · Reviewer_sY6f · 2026-04-01
> >
> > Thanks for the response which has fully addressed my concerns. Please consider include the additional experiments results into the final version of the paper. I will update my score to 5.

---

### Official Review · Reviewer_YzNK · 2026-03-12

**Soundness:** 2
**Presentation:** 2
**Significance:** 3
**Originality:** 3
**Overall Recommendation:** 3
**Confidence:** 3

**Summary:**

This paper proposes a graph-based perspective for analyzing hallucinations in LLMs. The core idea is to model next-token prediction as reasoning over an underlying directed graph, where nodes represent reasoning states and edges represent valid transitions. Under this framework, the paper distinguishes between intrinsic reasoning, which should be faithful to a context-induced subgraph, and extrinsic reasoning, which relies on knowledge stored in model parameters. The paper then identifies two failure modes: Path reuse, where the model reuses memorized paths and ignores context, and path compression, where multi-hop reasoning paths collapse into shortcut transitions during later training. The authors conduct synthetic graph experiments and a more realistic experiments to argue that these mechanisms help explain hallucination phenomena in LLMs.

**Compliance With Llm Reviewing Policy:**

Affirmed.

**Key Questions For Authors:**

- The shortest-path layers are defined as $V_i = { v \mid \mathrm{dist}(s,v) = i,; \mathrm{dist}(v,t) = d - i }$.
Since this definition plays an important role in the subsequent upper-bound argument, it would be helpful if the authors could explain why this (somewhat unusual) definition is used.

- The explanation of Path Compression is mostly qualitative. Could the authors provide a more formal justification or model explaining why this behavior emerges during training?

- The relationship between the underlying graph and token-level next-token prediction is not always clear. Could the authors clarify how reasoning paths correspond to the generated token sequences?

**Limitations:**

yes

**Strengths And Weaknesses:**

## Strengths

- **Core idea is interesting.**

  The paper offers a clean conceptual reframing of hallucinations as failures in graph-structured reasoning rather than only output-level factual errors. The distinction between intrinsic and extrinsic reasoning is useful.

- **Failure mode identification.**

  The idea that early hallucinations arise from memorized path reuse, while later ones emerge from shortcut compression, provides a simple and memorable explanation for different kinds of reasoning failures.

- **Simple experimental setup.**

  The synthetic graph setup is well designed for studying structural properties of reasoning behavior, and the experiments exploring graph density, communities, and node degree offer useful insights into how the model internalizes graph structure.

---

## Weaknesses
- **Limited empirical results.**
  Despite the interesting idea, the evidence supporting the central claims is still somewhat limited. Many of the conclusions are based on observations from synthetic graph environments, and it is not yet clear how strongly these mechanisms generalize to real-world language modeling settings.

- **Definition of 'hallucination'.**

  The paper uses the term 'hallucination' in a fairly broad sense. Some of the observed failures may be better interpreted as shortcut learning or reasoning-path errors rather than hallucinations in the traditional sense. A clearer discussion of how the proposed mechanisms relate to existing definitions of hallucination would help.

- **Limited empirical evidence.**

  Some claims about next token prediction as the root cause of these behaviors feel stronger than the current experiments demonstrate. More targeted ablations would be useful to support this claim.

Also, I found several writing issues that should be corrected for clarity.
- l.153: can be $2^{N+N^2}$ $\rightarrow$ $2^{\frac{N(N-1)}{2}}$?

  For a complete graph we have $|E| = \frac{N(N-1)}{2}$. Thus the number of possible edge subsets would typically be $2^{|E|} = 2^{\frac{N(N-1)}{2}}$.
- l.380 'making it continue for Transformers to achieve lower loss'
- l.431 'which reasoning under contextual constraints'

---

> ### Author Rebuttal · Authors · 2026-03-30
>
> Thanks for your insightful suggestions. We will add the discussions and details in our revision.
> >W1: Limited empirical results
>
> First, synthetic graphs provide a controlled setting to better understand LLM behaviors that are more complex in real-world data. Prior work has similarly used synthetic data to reveal model limitations [1][2]. Here, we assume full access to the search space and analyze LLM behavior within it, making this a predictive framework for identifying potential issues. Second, although LLMs are trained on natural language, real-world knowledge [3] and reasoning processes [4] can often be represented as graphs. In Section 6.1, we've shown how path compression during training on language can lead to hallucinations.
>
> Moreover, we will add two real-world experiments. First, we assume an internal knowledge graph from language. We use the DocRED dataset [5], which provides documents and annotated graphs. A 4-layer Qwen3 model is trained on the documents, and we obtain the uncompression ratio at epochs 20, 100, and 1000 are 0.47, 0.48, and 0.40. This shows a similar trend to synthetic graphs: increasing early, then decreasing later. Second, we analyze language data using community detection. We identify 71, 136, and 770 communities in GSM8K (Math), APPS (Code), and DocRED (KG), respectively. These results indicate clear community structure in language data, supporting our analysis.
> >W2: Definition of hallucination
>
> Building on prior work [6][7], we focus on two types of hallucinations [6]. For Input-Conflicting hallucinations, models rely on prior knowledge instead of the given input; we explain this as path reuse. For Fact-Conflicting hallucinations, models produce reasoning errors that contradict known facts; we describe this as path compression, a form of shortcut learning during training.
> >W3: Empirical evidence
>
> To investigate whether next-token prediction (NTP) is the core factor, we examine two alternative loss formulations that evaluate the impact of different training targets: a diffusion-based method (DLLM, using LLADA) and an edge-constrained variant (NTP-EC). We run experiments on a 500-node graph ($p_{\text{in}}=0.1$, $p_{\text{out}}=0.01$) and evaluate path prediction accuracy. The edge constraint is:$L_{\text{edge}}=1/(m-1)\sum_{i=1}^{m-1}\ell_{\text{BCE}}(s_i,y_i)$ where $m$ is path length, $s_i$ the predicted logit for consecutive token pairs, and $y_i$ indicates edge existence.
> |Epoch|0.5|1|1.5|5|15|25|30|40|
> |-|-|-|-|-|-|-|-|-|
> |NTP|0|0|0|.11|.87|.78|.77|.67|
> |DLLM|0|0|0|0|.46|.64|.49|.72|
> |NTP-EC|.16|.88|.98|1|1|1|1|1|
>
> NTP improves then degrades. DLLM is unstable later. NTP-EC quickly reaches and maintains high accuracy. Only NTP shows a consistent late-stage decline, supporting our hypothesis. More details will be added in the revision.
> > W4 & Q1: formulation clarity
>
> We would like to clarify our presentation as follows:
> - l. 153: We assume the underlying graph is directed. Therefore, the maximum number of edges is $2^{|E|}=2^{N(N-1)}$.
> - l.176: For the shortest-path case, we emphasize the middle-node term, as the local density between neighboring nodes (e.g., $v_i \to v_{i+1}$) determines the maximum number of possible paths.
> - l.380: In real-world corpora, auxiliary tokens introduce additional noise and interference. Thus, while the overall loss decreases, path compression remains observable at the knowledge graph level.
> - l.431: which relies on memorized underlying knowledge.
>
> > Q2 & Q3: the explanations
>
> We will provide a formal justification and proofs in the revision.
>
> First, by the optimal substructure of shortest paths, any suffix of an optimal path is also optimal. This motivates modeling autoregressive generation as local next-step prediction, $P_\theta(v_{k+1}|v_k)$, independent of earlier history.
> For underlying graph, we set the transition matrix for the nodes as $T_{xy}=1/\mathrm{outdeg}(x)$ and the model’s learned transition as a mixture of $i$-step random walks: $P_\theta(y| x)=\sum_{i=1}^K\lambda_i^\theta T^i_{xy}$ , where $\lambda_i^\theta$ are learned mixing weights.
> **Proposition:** Suppose in 2-hop case, there is a path $v\to m\to u$, and there is no direct edge $v\to u$. Let $\mathcal{M}$ be the set of intermediate nodes $v\to m'\to u$. If the learned weights satisfy: $\lambda_2^\theta/\lambda_1^\theta>1/(\sum_{m'\in\mathcal{M}}\frac{1}{\text{outdeg}(m')})$, then the model predicts the skipped node with higher probability than the true adjacent node: $P_\theta(u|v)>P_\theta(m|v)$.
>
> [1] Transformers Struggle to Learn to Search
> [2] The Pitfalls of Next-Token Prediction
> [3]Physics of Language Models: Part 1, Learning Hierarchical Language Structures
> [4]Topology of Reasoning: Understanding Large Reasoning Models through Reasoning Graph Properties
> [5]DocRED: A large-scale document-level relation extraction dataset
> [6]Survey of hallucination in natural language generation
> [7]Siren’s Song in the AI Ocean: A Survey on Hallucination in Large Language Models

---

> > ### Author Rebuttal · Reviewer_YzNK · 2026-04-01
> >
> > Thank you for the additional experiments and clarifications. I appreciate the effort to strengthen the paper. The new results are helpful, but I remain concerned that the paper's claims are still somewhat stronger than what the current evidence supports. Therefore, I will keep my score unchanged.

---

> > > ### Author Response · Authors · 2026-04-02
> > >
> > > Thank you for your prompt response, which allows us to further clarify our contributions and elaborate on how we conclude our findings.
> > >
> > > **Why observing in a graph environment**: The capabilities of language models depend on both the training data and the model architecture. However, a key challenge lies in the complex pretraining data and the high cost of training large-scale language models. As a result, simplifying both the training data and model architecture has become a widely adopted approach to explore the capability boundaries of decoder-only transformers[1][2]. Among them, graph-structured data, which is easily obtainable and well-defined, has also been adopted to study planning[3] and searching[4] abilities in LLMs.
> > >
> > > **Why explaining hallucinations in graph view**: In particular, we focus on the phenomenon of hallucination, examining two representative formulations: input conflict and fact conflict [5].
> > >
> > > Prior work defines input conflict as cases in which LLM outputs are not aligned with the provided input [5]. In our framework, this can be interpreted as follows: transformers first internalize an underlying graph structure of knowledge. Input conflict then arises when the model generates outputs that deviate from the subgraph specified by the input. Fact conflict, on the other hand, refers to cases in which model outputs are inconsistent with established world knowledge or cannot be verified. For example, when asked about the mother of Afonso II, an LLM might incorrectly answer “Queen Urraca of Castile” instead of the correct “Dulce Berenguer of Barcelona” [5].
> > >
> > > Both types of hallucination can be understood as manifestations of knowledge conflict. In particular, when user input itself constitutes a source of established world knowledge, it should be viewed as a subset of the broader knowledge space. Since knowledge in the sentences can be formulated as a knowledge graph (at the entity understanding level[6] or reasoning step level[7]), we introduce the concept of an underlying graph within our framework. Individual sentences can then be interpreted as samples drawn from this global knowledge structure.
> > >
> > > This perspective helps us analyze when and how knowledge conflicts arise during LLM pretraining. For example, with limited data or early in training, the model may retrieve paths not grounded in the input but in the underlying graph, which we call path reuse. In contrast, fact conflicts stem from path compression, which typically appears later in pretraining. As discussed in Section 6.1, sufficiently complex real-world data can trigger this behavior even before training loss fully converges.
> > >
> > > **How do we analyze the empirical results**: We investigate these phenomena from two complementary perspectives: data distribution and model architecture.
> > >
> > > We provide a comprehensive analysis of data distribution in the main paper. For instance, we observe that models tend to skip previously generated tokens under certain conditions, particularly when the underlying graph contains high out-degree nodes. We further analyze analogous phenomena in natural language settings (Sec 6.1) and incorporate additional real-world data analysis in the rebuttal.
> > >
> > > On the architectural side, we focus on decoder-only transformers and evaluate several representative models, including Qwen, Mixtral, and LLaMA. Through layer-wise analysis (Sec 5.2), we show that these models consistently exhibit failure modes under various conditions. Additionally, in the rebuttal, we analyze the target function and find that the diffusion-based formulation can be unstable, whereas incorporating edge constraints helps alleviate these issues. This suggests that reliance on next-token prediction alone may contribute to the observed behaviors.
> > >
> > > Taken together, our findings indicate that during pretraining, the interaction between high out-degree nodes in the data distribution and the next-token prediction objective can lead to path compression phenomena. However, this issue can be alleviated during post-training. As discussed in Section 6.2, reinforcement learning methods can help align model reasoning with the underlying knowledge structure.
> > >
> > > We will add additional discussion and clarify the relevant descriptions in the revision. In this work, we conduct experiments from multiple perspectives to support our analysis. If there are further aspects that could strengthen our claims, we would be happy to continue the discussion.
> > >
> > > [1] Physics of language models: Part 3.3, knowledge capacity scaling laws
> > >
> > > [2] Weight-sparse transformers have interpretable circuits
> > >
> > > [3] The Pitfalls of Next-Token Prediction
> > >
> > > [4] Transformers Struggle to Learn to Search
> > >
> > > [5]Siren’s Song in the AI Ocean: A Survey on Hallucination in Large Language Models
> > >
> > > [6]DocRED: A large-scale document-level relation extraction dataset
> > >
> > > [7]Topology of Reasoning: Understanding Large Reasoning Models through Reasoning Graph Properties

---

### Official Review · Reviewer_cCrD · 2026-03-13

**Soundness:** 3
**Presentation:** 4
**Significance:** 4
**Originality:** 3
**Overall Recommendation:** 5
**Confidence:** 3

**Summary:**

The authors develop a unified graph-based framework to explain the search behavior of transformers and analyze how intrinsic and extrinsic reasoning emerge. They further explain hallucinations in decoder-only models through two behaviors: path reuse and path compression. They also analyze how reasoning hallucinations relate to well-known LLM behaviors.

**Compliance With Llm Reviewing Policy:**

Affirmed.

**Final Justification:**

The rebuttal has reinforced my prior assessment. It was mainly a clarification question.

**Key Questions For Authors:**

W1 is my key main question.

**Limitations:**

Yes

**Strengths And Weaknesses:**

Strengths:
S1. Overall a well-written paper
S2. The research is well-motivated and experimented.

Weakness/Clarification:
W1. Based on the results in Figure 4, how can we disentangle whether the observed behaviors reflect genuine model behavior versus artifacts of the graph formulation imposed by the authors?

---

> ### Author Rebuttal · Authors · 2026-03-30
>
> > W1. Based on the results in Figure 4, how can we disentangle whether the observed behaviors reflect genuine model behavior versus artifacts of the graph formulation imposed by the authors?
>
> A: Thank you for your insightful question. Hallucination in LLMs has been discussed and defined in prior work [1][2]. In particular, Figure 4 illustrates fact-conflicting hallucinations [2], where graphs are used as a tool to simulate real-world data distributions and to better understand why hallucinations arise during the training process.
>
> This approach is motivated by the observation that graphs are a widely used abstraction for representing language-related structures. For example, knowledge encoded within languages can be viewed as forming structured representations, drawing on concepts such as knowledge graphs [3] and the topological structure of reasoning [4].
>
> Based on this perspective, we first construct a graph and then perform path sampling to simulate the idea that language follows a consistent logical structure. Within this framework, hallucination can be interpreted as an inherent behavior of the model, with the graph serving as a lens through which this phenomenon can be analyzed and better understood.
>
>
> [1]Survey of hallucination in natural language generation
>
> [2]Siren’s Song in the AI Ocean: A Survey on Hallucination in Large Language Models
>
> [3] DocRED: A large-scale document-level relation extraction dataset
>
> [4]Topology of Reasoning: Understanding Large Reasoning Models through Reasoning Graph Properties

---

> > ### Author Rebuttal · Reviewer_cCrD · 2026-04-04
> >
> > Thank you for your response! My clarification has been addressed and I will keep my positive assessment of the paper as well.

---

### Official Review · Reviewer_sBzs · 2026-03-21

**Soundness:** 2
**Presentation:** 4
**Significance:** 3
**Originality:** 3
**Overall Recommendation:** 5
**Confidence:** 2

**Summary:**

This paper looks at reasoning hallucinations in decoder-only Transformers from a graph perspective. The authors model next-token prediction as a path search process over an implicit directed graph. In this graph, nodes represent entities or reasoning states, and edges represent valid transitions. They distinguish between context-constrained (intrinsic) reasoning and memory-driven (extrinsic) reasoning.
The authors identify two main mechanisms behind hallucinations: Path Reuse occurs in early training when the model overly relies on memorized paths and ignores context constraints; Path Compression appears in later training when multi-hop reasoning paths collapse into shortcut edges. The paper provides evidence using synthetic graph tasks, examines training dynamics, proposes a simpler theoretical explanation, and validates the phenomena in natural language settings under various fine-tuning strategies.

**Compliance With Llm Reviewing Policy:**

Affirmed.

**Final Justification:**

My concerns are mainly addressed and therefore I remain my positive score.

**Key Questions For Authors:**

1. Could you report the actual time taken to train a model in the paper?
2. What is the relationship between your graph-based interpretation and former work that try to give interpretation from other perspectives?
3. It would be better if authors could define the critical concepts (e.g. hallucination) based on or extended from some established works.

**Limitations:**

No.

**Strengths And Weaknesses:**

**Strengths**

1. The core idea is intuitive and coherent. The proposed mechanisms (Path Reuse and Path Compression) align well with common intuitions about underfitting and overfitting in next-token prediction, offering a unified explanation for different types of hallucinations.

2. The experimental design is relatively comprehensive. The paper spans controlled synthetic settings, structural factor analysis, theoretical modeling, natural language validation, and fine-tuning experiments, forming a well-connected investigation pipeline.

3. The writing is clear and well-structured. The logical flow is easy to follow, and the distinction between different reasoning settings helps make the mechanisms understandable.

**Weaknesses**

1. The related work is not sufficiently comprehensive, especially regarding prior research on hallucination mechanisms and internal interpretability. The paper positions its contribution as a unified explanation of hallucination mechanisms, but there already exists a substantial body of work that studies hallucinations from internal-state, representation-level, or knowledge-conflict perspectives [1][2]. While the proposed graph-based and training-dynamics perspective is indeed novel, the current discussion does not clearly distinguish this work from existing approaches. As a result, the boundary of novelty is not sufficiently well articulated.

2. Some core concepts remain operational and not fully rigorous. Although the paper frames its objective as studying “reasoning hallucination,” the experimental setup primarily defines and measures two types of graph-level deviations: violations of contextual subgraph constraints and the generation of non-existent edges in the underlying graph. In this sense, the work more precisely characterizes graph-structured reasoning errors rather than general hallucination. Similarly, the intrinsic/extrinsic reasoning distinction holds under the controlled experimental setup, but real-world scenarios typically involve mixed interactions between parametric and contextual knowledge. Prior work has shown that these interactions can be supportive, complementary, or conflicting [3]. Therefore, the current formulation appears closer to two idealized endpoints, and may not naturally extend to more general reasoning settings.

3. The statistical reliability of the experimental results is not sufficiently established. Many of the key claims rely on training dynamics curves that appear to be based on single-run trajectories. At the same time, the paper itself reports instability and performance fluctuations in several metrics at later stages of training. Without repeated experiments across multiple random seeds and reporting of mean and variance, it is difficult to determine whether these trends reflect the proposed mechanisms or simply optimization noise near convergence. Although some hardware and hyperparameter details are provided, the lack of systematic multi-run evaluation weakens the empirical support for the claimed stage-dependent mechanisms. A more robust approach would be to first validate the findings on smaller models with multiple initializations and configurations, and then demonstrate that the trends generalize to larger architectures.

[1] PoLLMgraph: Unraveling Hallucinations in Large Language Models via State Transition Dynamics.

[2] LLMs Know More Than They Show: On the Intrinsic Representation of LLM Hallucinations.

[3] Understanding the Interplay between Parametric and Contextual Knowledge in LLMs.

---

> ### Author Rebuttal · Authors · 2026-03-30
>
> >W1 & Q2: Comparing with previous work
>
> A: Thanks for your valuable feedback. Our work differs from prior studies in its focus on pretraining stage and data distribution. We assume that LLMs exhibit hallucinations under certain data distributions shaped during pre-training, suggesting that hallucination is a dynamically evolving phenomenon. In contrast, most existing work examines hallucinations when pre-training has been completed. For example, PoLLMgraph analyzes hidden state distributions [1], while [2] investigates representations within specific layers of a fully trained model at a fixed pretraining stage. These approaches provide important insights into internal representations but do not account for how such representations develop over time. Our work emphasizes the evolution of models during pre-training, whereas prior studies primarily analyze internal representations in fully trained models. We believe that bridging these two perspectives is important, and we will incorporate this discussion into the revision.
>
> > W2: Some core concepts remain operational and not fully rigorous.
>
> A: Thank you for your questions. We use graphs as a simplified, controllable setting to study hallucination. In practice, real-world knowledge[4] and reasoning topology[5] inside LLMs are often graph-structured, and prior work has already used graphs to analyze LLM reasoning behaviors[6][7]. Thus, we do not introduce specific graph-level designs for graph-structured reasoning.
>
> While real-world scenarios are more complex, our setup can explain many observed phenomena such as these noted in [3]. For instance, knowledge tied to high out-degree nodes is more likely to be recalled, reflecting dataset imbalance. We also observe that when training data covers only 0.1% of the search space, intrinsic knowledge degrades in later reasoning stages, which may explain reduced confidence in internal knowledge.
> >W3&Q1: multi-turn run and actual time to train.
>
> A: Thanks for your feedback. We provide the multi-turn and runtime results in the table below. For each graph setting, we run three different random seeds and report the mean and standard deviation of the accuracy. In addition, we report the minimum runtime observed across the three seeds.
>
> The results are consistent across different graph settings and random seeds at each epoch, exhibiting a trend that first increases and then decreases, indicating stable path compression behavior. We will revise the figures accordingly and include detailed runtime results in the revised version.
>
> | #community_$p_{in}$_$p_{out}$ | epoch-1        | epoch-7        | epoch-49       | time (ms) |
> |----------------------|----------------|----------------|----------------|-----------|
> | 10_0.3_0.01          |  0.992±0.00   | 0.998±0.00     | 0.983±0.00   | 1197.1    |
> | 10_0.6_0.01          | 0.998±0.00     | 0.999±0.00     | 0.997±0.00     | 1805.54   |
> | 50_0.3_0.01          | 0.904±0.02     | 0.979±0.04     | 0.888±0.14     | 324.62    |
> | 50_0.6_0.01          | 0.967±0.07     | 0.978±0.02     | 0.898±0.03     | 375.81    |
> | 100_0.1_0.01         | 0.952±0.01    | 0.959±0.01    | 0.851±0.00    | 199.79    |
> | 100_0.1_0.02         |  0.962±0.002   | 0.970±0.003     | 0.831±0.011     | 393.95    |
>
> >Q3: It would be better if authors could define the critical concepts (e.g. hallucination) based on or extended from some established works.
>
> A: Thank you for your advice. Our work aims to explain current hallucination behaviors in LLMs. We interpret Input-Conflicting hallucinations[5] as path reuse, where LLMs rely on prior knowledge rather than faithfully following the input context. In contrast, Fact-Conflicting hallucinations[5] are explained by path compression, where important intermediate nodes are omitted, leading to unfaithful outputs. We will clarify these points in the revision.
>
> [1] PoLLMgraph: Unraveling Hallucinations in Large Language Models via State Transition Dynamics.
>
> [2] LLMs Know More Than They Show: On the Intrinsic Representation of LLM Hallucinations.
>
> [3] Understanding the Interplay between Parametric and Contextual Knowledge in LLMs.
>
> [4] Survey of hallucination in natural language generation
>
> [5]Siren’s Song in the AI Ocean: A Survey on Hallucination in Large Language Models
>
> [6] Transformers Struggle to Learn to Search
>
> [7] The Pitfalls of Next-Token Prediction

---

> > ### Author Rebuttal · Reviewer_sBzs · 2026-04-01
> >
> > My problems are largely addressed, thank you. I will remain my positive assessment.

---

### Decision · Program_Chairs · 2026-04-30

**Decision:**

Accept (regular)

**Comment:**

The paper proposes a graph-based framework for understanding reasoning hallucinations in decoder-only language models, introducing the notions of path reuse and path compression as two failure modes that arise during training. Overall, the reviewers agreed that the paper is interesting, well motivated, and clearly written, and several found the graph perspective to be a creative and useful way to study reasoning behavior in a controlled setting. The positive reviews emphasized the coherence of the framework, the breadth of the investigation across synthetic and more realistic settings, and the paper’s ability to connect its proposed mechanisms to known empirical phenomena in LLM reasoning.

The less supportive reviews focused mainly on the extent to which the synthetic graph findings can be carried over to real natural language settings, and on whether graph-level deviations should be interpreted as hallucinations in the broader sense. These are reasonable concerns, but in my reading the rebuttal addressed them substantially: the authors added experiments on real knowledge graphs and language data, clarified the intended scope of the claims, provided multi-seed results, and strengthened the discussion connecting the controlled graph setting to natural language structure. Importantly, even the less supportive reviewers appear to have softened some of their concerns, with one lowering the severity of their criticism to a weak accept after the rebuttal. Overall, I find the paper’s central idea compelling, the empirical story sufficiently strengthened by the rebuttal, and the remaining limitations appropriate to acknowledge rather than disqualifying. I am therefore supportive of the paper.